

# The mediating roles of coping styles and academic burnout in the relationship between stressors and depressive symptoms among Chinese postgraduates

Hong Shi[1], Hanfang Zhao[1], Minfu He[1], Zheng Ren[1], Shixun Wang[1], Li Cui[1], Jieyu Zhao[1], Wenjun Li[1], Yachen Wei[1], Wenjing Zhang[1], Ziqiang Chen[1], Hongjian Liu[2] and Xiumin Zhang[1]

[1] Department of Social Medicine and Health Management, Jilin University School of Public Health, Changchun, China

[2] Department of Epidemiology and Biostatistics, Jilin University School of Public Health, Changchun, China

## ABSTRACT

**Background**. Since few studies have incorporated factors like stressors, coping styles, and academic burnout into the same model to analyze their impacts on depressive symptoms, this research attempts to establish an optimal structural model to explore the direct and indirect effects of these factors on depressive symptoms.

**Methods**. A total of 266 postgraduates completed questionnaires regarding coping styles, academic burnout, stressors, and depressive symptoms. The path analysis was applied for investigating the roles of coping styles and academic burnout in mediating the relationship between stressors and depressive symptoms.

**Results**. The total and direct effects of stressors on depressive symptoms were 0.53 and 0.31, respectively. The proportion of the direct effect of stressors on depressive symptoms to its total effect amounted to 58.50%. The indirect effects of academic burnout, positive coping style, and negative coping style on the association between stressors and depressive symptoms were 0.11, 0.04, and 0.03, taking up 20.75%, 7.55%, and 5.66% of the total effect, respectively. The serial indirect effect of positive coping style and academic burnout was 0.02, accounting for 3.77% of the total effect, while that of negative coping style and academic burnout was 0.02, taking up 3.77% of the total effect.

**Conclusions**. Coping styles and academic burnout chain jointly mediate the relationship between stressors and depressive symptoms among postgraduates. Thus, encouraging postgraduates to tackle stress positively may reduce the likelihood of the development of academic burnout and further reduce depressive symptoms.

Corresponding author
Xiumin Zhang,
zhangxiumin63@163.com

## INTRODUCTION

Postgraduate education directly determines the quality of national talent training (*Tang, 2022*). As China expands college and university enrollment, the population of postgraduates

in China has been rising (*Tang, 2022*). In their everyday life, they are constantly under heavy stress and thus subject to a high risk of mental disorders. Moreover, more and more suicides have been reported among postgraduates (*Too et al., 2019*; *Cheng et al., 2020b*). As revealed by a report on National Mental Health Development in China (2019–2020), 35.50% of Chinese postgraduates might have depressive symptoms (*Fu, Zhang & Chen, 2021*). Also, according to a comprehensive survey of 2, 279 individuals from 26 countries and 234 institutions, postgraduates were six times more likely to experience depression than the general population (*Evans et al., 2018*). Furthermore, a study on the mental health of American graduate students indicated that American graduate students were twice more likely to suffer moderate or severe levels of depression or anxiety than the general population (*Bolotnyy, Basilico & Barreira, 2022*). According to the results of a meta-analysis, over 33.00% of the investigated postgraduates mentioned depressive symptoms, indicating that they are vulnerable to mental health threats (*Guo et al., 2021*). These studies have demonstrated that the mental health of higher education students is not optimistic. This highlights the necessity to pay more attention to depressive symptoms among postgraduates. In the meantime, understanding the related factors and internal path of depressive symptoms is critical for the development of effective intervention measures.

Stress is prevalent among postgraduates. The report on National Mental Health Development in China (2019–2020) indicated that about 70.00% of postgraduates worked 8 hours or more per day and 36.50% worked 10 hours or more per day. Heavy academic burden, unsatisfactory employment prospects, and obscure self-positioning are the major sources of stress for postgraduates (*Fu, Zhang & Chen, 2021*). According to the stress process model, stressors affect health. This model involves three aspects of stress: stress sources (*e.g.*, chronic stressors and life events), stress mediators (*e.g.*, self-concepts, social support, and coping skills), and stress outcomes (*e.g.*, physical and mental health problems) (*Pearlin, 1989*). Compared with acute stressors, chronic stressors can predict depressive symptoms more accurately (*Mcgonagle & Kessler, 1990*). Having been exposed to a competitive and stressful environment for a long time, postgraduates are more subject to depressive symptoms. Previous research has supported the positive association between stress and depressive symptoms among junior high school students, college students, medical residents, medical postgraduates, and biomedical doctoral students (*Acharya, Jin & Collins, 2018*; *Hish et al., 2019*; *Hyakutake et al., 2016*; *Shah et al., 2021*; *Zhu & Li, 2016*). In addition, depressive symptoms can be predicted by academic stressors, family and monetary stressors, workload and time management stressors, *etc.* (*Jones-White et al., 2022*; *Hish et al., 2019*; *Saravanan & Wilks, 2014*).

In addition to depression symptoms, another consequence of long-term exposure to stressors is burnout (*Kumar, 2016*). According to the demands-resources model, such burnout can be a result of the imbalance between demand and resource. More specifically, demand overload can drain individuals' energy and lead to burnout (*Demerouti et al., 2001*). In the school setting, students who have been facing academic stress and demand overload for a long time may lose their interest in learning and get exhausted (*Jiang et al., 2021*; *Salmela-Aro & Upadyaya, 2014*). Recognized as an extension of burnout, academic

burnout is referred to as negative attitudes, learning mindset, and behaviors toward study, which is caused by stress and a lack of learning motivation (*Wang et al., 2020*; *Zhang, Gan & Cham, 2007*). Academic burnout has three dimensions: dejection, improper behavior, and reduced personal accomplishments (*Lian, Yang & Wu, 2005*; *Wang et al., 2021*). Previous studies have shown the prevalence and urgency of academic burnout among college students, as well as its adverse impacts on one's personal development (*Chunming et al., 2017*; *Wang et al., 2020*). Nevertheless, little research attention has been paid to academic burnout in the postgraduate population.

Academic burnout is a unique stress outcome in academia when students are confronted with stressors. Academic burnout and the associated depressive symptoms are crucial concerns related to the well-being of postgraduates. Compared with depressive symptoms, burnout symptoms tend to be more common (*Colville & Smith, 2017*). According to a study on Chinese neurology graduate students, burnout may be considered a state prior to depression symptoms. It usually appears alone, while depressive symptoms do not (*Zhou et al., 2021*). Furthermore, as confirmed in a previous study, academic burnout is associated with negative psychological health (*Jiang et al., 2021*). While academic burnout may directly lead to depressive symptoms, empirical studies have demonstrated that stress may also be associated with one's academic burnout. This implies the potential mediating role of academic burnout in the relationship between stress and depressive symptoms. Therefore, in this study, it is hypothesized (H1) that stressors are positively correlated with depressive symptoms indirectly by academic burnout.

In order to help postgraduates cope with stress effectively and thus reduce academic burnout and depression symptoms caused by stressors, it is significant to explore effective interventions. Coping means that one makes conscious efforts to regulate cognition, behavior, physiology, emotion, and the environment in response to stress (*Compas et al., 2001*). As an individual behavioral and cognitive strategy, coping can be grouped into two categories when a stressor is perceived: positive coping styles (including attempting to change and seeking support from others) and negative coping styles (including venting and avoidance) (*Compas, Orosan & Grant, 1993*; *Li et al., 2020*). The stress process model indicates that coping skills as stress mediators can affect stress outcomes (*Pearlin, 1989*). Adverse experiences can exert an impact on the development of individuals' coping styles. For example, adolescents who face more adversities tend to employ more maladaptive coping strategies and fewer adaptive ones (*Arslan, 2017*). Moreover, it has been noted that active coping strategies and negative ones partially and fully mediate the relationship between psychological maltreatment and internalized and externalized problems among adolescents, respectively (*Arslan, 2017*). A previous study also suggests that emotion-oriented coping partially mediates the relationship between stress and disordered eating among adolescents. Individuals experiencing a high level of stress tend to adopt an emotion-oriented coping style, which may result in a higher level of disordered eating (*Henderson et al., 2022*). These empirical studies have confirmed that a high level of perceived stress is associated with a higher frequency of employing negative coping strategies, which intensify mental or behavioral health problems. If the positive coping style is adopted in response to stressors, it can be regarded as a factor favorable for mental or behavioral health. On

the contrary, the negative coping style can be deemed a factor detrimental to mental or behavioral health. Postgraduate students can hardly avoid the strain derived from tasks such as writing and publishing papers and social practice because these stressors are vital to postgraduate education in the training environment. In the meantime, postgraduates have varying abilities to interpret and tackle stressors when facing them. Therefore, their reactions and coping styles are also likely to be different. The individuals' appraisal of the stress situation will affect the coping styles they use, which, in turn, affects the development of symptoms (*Henderson et al., 2022*). Moreover, individuals' coping styles explain the different outcomes caused by the same stressor in different populations. For example, a positive coping style usually results in a lower risk of depressive symptoms (*Li, 2022*). To prevent the occurrence of more stress outcomes among postgraduates, it is feasible to develop a positive coping style. Based on empirical studies that have evidenced the association between stressors and coping styles, it can be expected that postgraduates can adopt a more negative or less positive coping style when perceiving higher stress or more stressors. More stressors are associated with a higher frequency of using negative coping styles, which promotes the development of negative stress outcomes. To sum up, according to the stress process model and previous empirical research findings, the study ideas are as follows: Stressors (postgraduate daily stressors) → Stress mediators (coping styles) → Stress outcomes in academia (academic burnout) → Stress outcomes in mental health (depressive symptoms). Accordingly, it is hypothesized that both the positive coping style (H2) and the negative coping style (H3) mediate the relationship between stressors and depressive symptoms. Furthermore, a positive coping style and academic burnout chain (H4), as well as a negative coping style and academic burnout chain (H5), mediate this relationship.

As highlighted before, postgraduates are an understudied group who receive training in a highly stressful and competitive environment. Moreover, few studies have incorporated coping styles, academic burnout, and stressors into the same model to analyze their relationship with depressive symptoms in this population. In view of this, the current study attempts to develop an optimal structural model for the relations between depressive symptoms and the aforementioned factors and analyze the direct and indirect effects of these factors on depressive symptoms. By exploring the mediating roles of coping styles and academic burnout in the relationship between stressors and depressive symptoms among Chinese postgraduates, it aims to provide evidence and suggestions for the development of educational interventions targeted at postgraduates and improve their mental health. Figure 1 shows the theoretical model.

## MATERIALS & METHODS

### Study design and subjects

In the cross-sectional design, the postgraduates from a university in Jilin Province of China who volunteered to participate in this survey were selected as the study participants in October 2022 by convenience sampling. They were asked to fill in a paper-based questionnaire. In the self-reported questionnaire, the participants' demographic information and their perceptions of stressors, coping styles, academic burnout, and

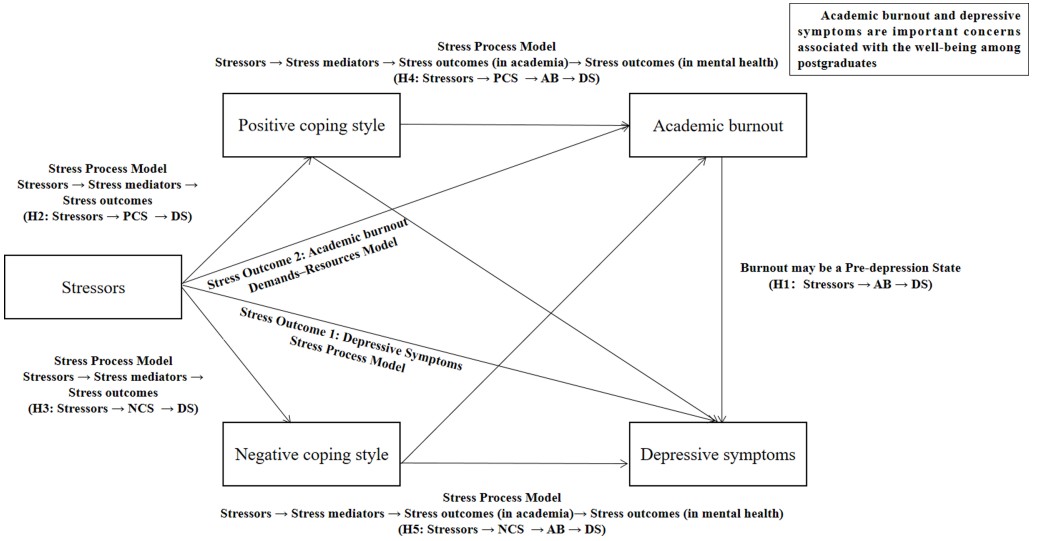

**Figure 1** Hypothesized mediating roles of coping styles and academic burnout between stressors and depressive symptoms.

depressive symptoms were provided. In this study, 267 respondents participated, of whom 266 finished the questionnaire. This study was approved by the Research Ethical Committee of Jilin University School of Public Health (Approval number: 2022-06-07). All participants were informed of the goals and significance of the survey and signed a written informed consent prior to the survey.

## Measures
### Demographic characteristics
Demographic characteristics cover gender (male/female), age (year), registered residence (urban/rural), the only child (Yes/No), average monthly household income (≤3000/3001-5000/>5000), and average monthly living expenses (≤1500/1501-2000/>2000).

### Postgraduate daily stressors scale
The Postgraduate Daily Stressors Scale was applied to measure the severity of stressful life events among postgraduates (*Huang, 2006*), including 31 items covering six domains of stressors: economic stressors, academic stressors, employment stressors, change stressors, love and sexual stressors, and social and other stressors. The scale quantified the occurrence of major life events that might occur in postgraduates' life and their impacts on individuals, and the total score was taken as the index of the psychological stress degree. Based on the occurrence of life events, the scale was assigned a score ranging from 1 to 5. 0 means the event did not occur, and the score increased as the influence of the event on individuals became stronger. The higher the score was, the greater the impact of postgraduates' perceived stressors would be. In this study, Cronbach's alpha was 0.91 for the Postgraduate Daily Stressors Scale.

### Simplified coping style questionnaire (SCSQ)

For the measurement of the participants' coping styles, the Simplified Coping Style Questionnaire (SCSQ) was adopted. In line with the characteristics of Chinese people, the scale was designed with satisfactory validity and reliability (*Xie, 1998*), including 20 items and two coping styles: items 1–12 are positive ones, and the rest are negative ones. There are four options, namely "never", "sometimes", "often", and "almost always", each of which was assigned a score (0, 1, 2, and 3 respectively). Thus, in each style, the mean of each item could be calculated. The higher the score was, the higher the frequency of using the coping style would be. In this study, Cronbach's alphas were 0.80 for the positive coping style and 0.73 for the negative coping style.

### Postgraduate academic burnout scale

The Postgraduate Academic Burnout Scale was employed to evaluate the level of academic burnout among postgraduates (*Cheng, Li & Sang, 2008*; *Li, 2009*). Based on the Academic Burnout Scale (*Lian, Yang & Wu, 2005*), the Postgraduate Academic Burnout Scale was designed by Chinese researchers with 20 items involving three domains of burnout. There were eight items about dejection, five about improper behavior, and seven on reduced personal accomplishments. Among them, eight items were reversed scoring items. A five-point Likert scale was used, ranging from 1 (totally disagree) to 5 (totally agree). The higher the scores were, the higher the levels of academic burnout would be. In this study, Cronbach's alpha in the Postgraduate Academic Burnout Scale and its dimensions of dejection, improper behavior, and reduced personal accomplishments were 0.90, 0.81, 0.76, and 0.78, respectively.

### Self-rating depression scale (SDS)

In terms of the evaluation of depressive symptoms, the Chinese version of the Zung Self-Rating Depression Scale (SDS) was adopted (*Zung, 1965*). The scale was composed of 20 items, each of which was scored from 1 to 4. Among them, 10 gained positive scores, while the other 10 achieved negative ones. The gross score was the sum of the scores of 20 items. Then, the gross was multiplied by 1.25, and the integer part of the result was the standard score. Higher standard scores indicated more severe depressive symptoms. In this study, Cronbach's alpha was 0.85 for the SDS.

## Statistical analysis

Descriptive statistics were applied to examine the demographic characteristics of the participants, including gender, age, registered residence, the only child, average monthly household income, and average monthly living expenses. Stressors, positive coping style, negative coping style, academic burnout, and depressive symptoms were described using descriptive statistical analysis. Number (proportions), means, and standard deviations (SDs) were utilized to describe categorical and continuous variables, respectively. Moreover, the Kolmogorov–Smirnov test was conducted to analyze the normal distribution of stressors, coping styles, academic burnout, and depressive symptoms. Also, the skewness and kurtosis of the above-mentioned continuous variables were examined. The absolute values of skewness and kurtosis were less than 2 and 7, respectively, which approximately

conform to a normal distribution (*Curran, West & Finch, 1996*; *Kim, 2013*). Based on unadjusted and adjusted potential confounding factors, the associations of stressors, two coping styles, and academic burnout with depressive symptoms were examined through linear regression analysis. The hypothesized relationships were further examined by adopting the path analysis using bootstrap maximum likelihood estimation in the statistical package AMOS 23.0. Every $p$-value was two-sided and statistically significant when it was below 0.05. To perform these analyses, Statistical Product and Service Solutions (SPSS) 24.0 and AMOS 23.0 (IBM Corp, Armonk, NY, USA) were used.

# RESULTS

## Descriptive statistics of sample characteristics

The sample consisted of 266 Chinese postgraduates aged 24.97 on average, with a standard deviation of 1.87. Among them, 76.30% were females, 65.40% were urban residents and 47.70% were the only child in the family. Besides, about 31.60% of them had an average monthly household income of above 5,001 yuan, and around 53.70% of them had average monthly living expenses below 1,500 yuan. The demographic characteristics of the participants are illustrated in detail in Table 1. The mean scores of stressors, positive coping style, negative coping style, academic burnout, and depressive symptoms were 25.10 ($SD = 16.33$), 26.71 ($SD = 4.78$), 9.86 ($SD = 4.16$), 50.82 ($SD = 10.25$), and 40.85 ($SD = 9.05$) respectively. The distribution of the six domains of stressors among the postgraduates is as follows: 75.90% of them perceived economic stressors, 91.70% perceived academic stressors, 91.70% perceived employment stressors, 85.30% perceived change stressors, 75.60% perceived love and sexual stressors, and 85.30% perceived social and other stressors.

## Stressors, positive coping style, negative coping style, and academic burnout as predictors of depressive symptoms

According to the results of the Kolmogorov–Smirnov test, coping styles ($P = 0.200$) and academic burnout ($P = 0.200$) displayed a normal distribution, whereas stressors ($P < 0.001$) and depressive symptoms ($P = 0.002$) did not. Moreover, stressors (Skewness coefficient = 0.84, kurtosis coefficient = 0.42) and depressive symptoms (Skewness coefficient = 0.57, kurtosis coefficient = 0.06) were found to show approximately a normal distribution. The results of the linear regression analysis of stressors, coping styles, academic burnout, and depressive symptoms are presented in Table 2. It can be seen that stressors, positive coping style, negative coping style, and academic burnout were significantly related to depressive symptoms before the control of potential confounding factors ($\beta = 0.30$, $P < 0.001$; $\beta = -0.20$, $P < 0.001$; $\beta = 0.13$, $P = 0.012$; $\beta = 0.33$, $P < 0.001$). Furthermore, stressors, positive coping style, negative coping style, and academic burnout were still associated with depressive symptoms when gender, age, registered residence, the only child, average monthly household income, and average monthly living expenses were added to the model as covariates ($\beta = 0.31$, $P < 0.001$; $\beta = -0.20$, $P < 0.001$; $\beta = 0.13$, $P = 0.011$; $\beta = 0.32$, $P < 0.001$).

**Table 1 Demographic characteristics of participants (N = 266).**

| Variables | | N | Percentage or Mean (SD) |
|---|---|---|---|
| Gender | | | |
| | Male | 63 | 23.70 |
| | Female | 203 | 76.30 |
| Age (year) | | 266 | 24.97 ±1.87 |
| Registered residence | | | |
| | Urban | 174 | 65.40 |
| | Rural | 92 | 34.60 |
| The only child | | | |
| | Yes | 127 | 47.70 |
| | No | 139 | 52.30 |
| Average monthly household income (yuan) | | | |
| | ≤3,000 | 84 | 31.60 |
| | 3,001–5,000 | 98 | 36.80 |
| | ≥5,001 | 84 | 31.60 |
| Average monthly living expenses (yuan) | | | |
| | ≤1,500 | 143 | 53.70 |
| | 1,501–2,000 | 68 | 25.60 |
| | ≥2,001 | 55 | 20.70 |

**Table 2 Stressors, positive coping style, negative coping style and academic burnout as predictors for depressive symptoms.**

| Variables | Unadjusted | | P | Adjusted[a] | | P |
|---|---|---|---|---|---|---|
| | $\beta$ | t | | $\beta$ | t | |
| Stressors | 0.30 | 5.87 | <0.001 | 0.31 | 5.77 | <0.001 |
| Positive coping style | −0.20 | −4.07 | <0.001 | −0.20 | −4.02 | <0.001 |
| Negative coping style | 0.13 | 2.54 | 0.012 | 0.13 | 2.56 | 0.011 |
| Academic burnout | 0.33 | 5.86 | <0.001 | 0.32 | 5.66 | <0.001 |

**Notes.**
[a] Adjusted for gender, age (year), registered residence, the only child, average monthly household income (yuan), average monthly living expenses (yuan); $\beta$, standardized coefficients.

## Test of the study model

To fit the data, the maximum likelihood method was applied. The results were adjusted based on the variables of gender, age, registered residence, the only child, average monthly household income, and average monthly living expenses. After the theoretical model was adjusted according to model fit indices, the final output model is illustrated in Fig. 2. The results of the path analysis revealed the significance of all paths in the model. The research hypotheses H1, H2, H3, H4 and H5 were evidenced. The model had an acceptable fit (RMSEA, GFI, AGFI, NFI, IFI, TLI, CFI, and PNFI were 0.05, 0.97, 0.93, 0.91, 0.97, 0.94, 0.96, and 0.51, respectively), and their respective reference values were satisfied by all the fit indices. The path coefficient results showed that stressors were negatively associated with positive coping style ($\beta = -0.21$) but positively associated with
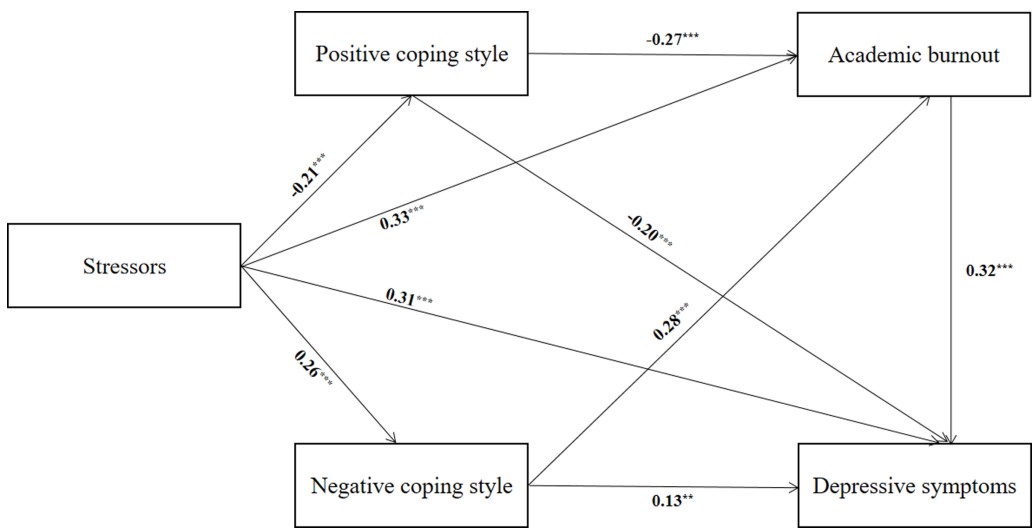

**Figure 2** **The final model and standardized model path.** All path coefficients were standardized.** $P <$ 0.01, *** $P < 0.001$.

**Table 3** **Direct and indirect effects of stressors on depressive symptoms.**

| | Path | Effect | Accounting for total effect, % |
|---|---|---|---|
| Direct effect | Stressors → Depressive symptoms | 0.31 | 58.50 |
| Indirect effects | 1. Stressors → Academic burnout → Depressive symptoms | $0.33 \times 0.32 = 0.11$ | 20.75 |
| | 2. Stressors → Positive coping style → Depressive symptoms | $(-0.21) \times (-0.20) = 0.04$ | 7.55 |
| | 3. Stressors → Negative coping style → Depressive symptoms | $0.26 \times 0.13 = 0.03$ | 5.66 |
| | 4. Stressors → Positive coping style → Academic burnout → Depressive symptoms | $(-0.21) \times (-0.27) \times 0.32 = 0.02$ | 3.77 |
| | 5. Stressors → Negative coping style → Academic burnout → Depressive symptoms | $0.26 \times 0.28 \times 0.32 = 0.02$ | 3.77 |
| Total effect | | 0.53 | ____ |

negative coping style ($\beta = 0.26$), academic burnout ($\beta = 0.33$), and depressive symptoms ($\beta = 0.31$). Moreover, positive coping style was negatively associated with academic burnout ($\beta = -0.27$), whereas negative coping style was positively associated with academic burnout ($\beta = 0.28$). Positive coping style was negatively associated with depressive symptoms ($\beta = -0.20$), whereas negative coping style was positively associated with depressive symptoms ($\beta = 0.13$). Academic burnout was positively associated with depressive symptoms ($\beta = 0.32$) (see Table S1). The details on the direct and indirect associations between stressors and depressive symptoms are presented in Table 3.

## DISCUSSION

Supporting postgraduates' mental health is a highly complicated issue, and universities and graduate programs have recognized the importance of initiatives that nourish postgraduates' mental health (*Hermanstyne et al., 2022*). In particular, the problem of depressive symptoms among postgraduates should be brought to the forefront (*Evans et al., 2018*). Concerning its complex nature, predicting depressive symptoms is challenging. To design effective interventions to alleviate depressive symptoms, the very first step should be clarifying the factors of depressive symptoms and how they work. The current study tried to probe into the mediating roles of coping styles and academic burnout between stressors and depressive symptoms among Chinese postgraduates through testing theoretical models inspired by the stress process model. In this way, the findings presented here extend the existing literature, thereby shedding new light on the paths between the above four variables and further explaining the stress-outcome relationships. Moreover, the findings of the current study enrich knowledge about depressive symptoms and the prevention of them among postgraduates.

Academic burnout would mediate the relationship between stressors and depressive symptoms. This finding supported the research hypothesis (H1). For one thing, stressors were positively associated with depressive symptoms among postgraduates, indicating that reducing stressors among postgraduates may promote a decline in their depressive symptoms. Moreover, stressors contributed to an increase in academic burnout among postgraduates, which may give rise to depressive symptoms. In particular, the findings of this study indicated that academic, employment, change, and social and other stressors are the main stressors for postgraduates. Among them, 91.70% perceived academic stressors, which is closely associated with academic burnout among postgraduates. In a highly stressful environment, postgraduates tend to perceive exhaustion, hold a cynical and indifferent study attitude, and feel incompetent. Previous research has pointed out that excessive workload, difficulties in learning and time management, work-life balance conflicts, peer relationships, and financial stressors are all potential problems that students may encounter (*Hill, Goicochea & Merlo, 2018*). Moreover, an existing study has manifested that students' academic burnout can be predicted by self-identity stress, interpersonal stress, future development stress, and academic stress (*Lin & Huang, 2014*). Meanwhile, academic burnout negatively affected students' academic performance and personal development (*Kong et al., 2021*; *Anderson et al., 2020*). Additionally, prolonged academic burnout would affect the sense of achievement and happiness and then elevate the risk of depressive symptoms (*Qian, Yin & Zhang, 2015*; *Wang et al., 2018*). This is in line with the previous research result that academic burnout might be a precipitating factor for depressive symptoms (*Cheng et al., 2020a*). Therefore, maintaining a relatively low level of perceived stress can benefit postgraduates' learning and life experience. It is necessary to reduce unnecessary stressors, and thus to alleviate academic burnout and eventually reduce depressive symptoms and promote psychological health.

This study supports the stress process model by offering evidence of the role of coping styles in mediating the relationship between stressors and depressive symptoms among

postgraduates. Stressors indirectly affected depressive symptoms through the simple mediating roles of positive coping style and negative coping style, respectively. The findings validated research hypotheses H2 and H3. A negative correlation between positive coping style and depressive symptoms was also identified, revealing the important prevention effect of positive coping style on depressive symptoms. This aligns with the findings of previous reports (*He & Li, 2021*; *Li et al., 2019*). Possible explanations are that postgraduates who frequently adopt positive coping styles are more adaptable to stressful environments and life changes. Postgraduates' positive thinking and attitude toward stressors can reduce the negative impacts of stressors to prevent or alleviate depressive symptoms. Therefore, interventions that encourage positive coping styles may be conducive to the prevention and reduction of depressive symptoms among postgraduates, especially for those faced with chronic stressors. The negative coping style mediated the relationship between stressors and depressive symptoms among postgraduates. This is in line with the result of prior research, that is, the negative coping style mediated the relationship between the COVID-19 pandemic-related stress and depressive symptoms among international medical students (*Lu et al., 2022*). The findings of the current research indicated that the postgraduates who perceived more stressors were more likely to employ more negative coping strategies. Individuals could not deal with stressful events correctly by adopting the negative coping style, which might even strengthen the inactive influence of stressors on them. Moreover, negative coping strategies, such as self-blame, escape, and fantasy adopted by individuals who were facing stressful events might increase their subjective experience of stressors, thus facilitating the development of depressive symptoms. In this way, reducing the frequency of adopting the negative coping style can indirectly lessen the adverse effects of daily stressors on depressive symptoms among postgraduates.

Additionally, the current study empirically examined the impact of the chain mediating roles of coping styles and academic burnout in the relationship between stressors and depressive symptoms. Thus, the two hypotheses were supported: Hypothesis H4 about the chain mediating effects of positive coping style and academic burnout on the relationship between stressors and depressive symptoms; Hypothesis H5 about the chain mediating effects of negative coping style and academic burnout on the relationship between stressors and depressive symptoms. The findings indicated the chain mediating roles of coping styles and academic burnout in explaining why postgraduates under stressors are more likely to have depressive symptoms. Since stressors cannot be eliminated, academic burnout and depressive symptoms, as two outcomes of stressors, may affect postgraduates' mental health and personal development. In a previous study, coping styles have been found to be potential mediating factors of academic burnout (*Wang et al., 2020*). Positive coping strategies, such as social support from family and friends, relaxation exercises, and sports had a negative association with academic burnout (*Erschens et al., 2018*). The positive coping style, as a protective factor, could alleviate the negative impacts of stressors on academic burnout. Conversely, negative coping strategies, such as alcohol use, drug use, and gambling on mobile phones were positively correlated with academic burnout (*Erschens et al., 2018*). Postgraduates adopting negative coping styles might be vulnerable to exposure to stressors, thus generating negative emotions and outcomes. The stressed postgraduates

who employed more negative coping styles were more subject to academic burnout, which could further exacerbate depressive symptoms. This may be a result of the weakened ability to find positive solutions to handle various stressors in stressful circumstances. By contrast, the active promotion of positive coping styles among stressed postgraduates might help to protect them from academic burnout, thereby reducing depressive symptoms. In this respect, developing positive coping skills and reducing academic burnout may be intervention strategies to prevent or lower depressive symptoms among postgraduates, especially for those under chronic stressors.

## Implications

Preventing and reducing depressive symptoms among postgraduates is part of comprehensive educational interventions. The findings of the current study have some implications for higher education. Targeted interventions include relieving stress, cultivating a positive coping style, and reducing academic burnout. For one thing, the negative emotions produced by stressors may be detrimental and should receive specific attention in postgraduate education. In terms of educators, they should help to avoid introducing excessive stress and lead postgraduates in setting appropriate academic goals. This can cut down academic burnout and further reduce depressive symptoms. Additionally, since stressors can hardly be eliminated, postgraduates should hold a more positive view to embrace the problems in life and study. Meanwhile, schools must provide relevant counseling programs to help students tackle stressors and reduce academic burnout and depressive symptoms.

For another, given the positive roles of positive coping skills in preventing and reducing adverse health outcomes, such skills should be prioritized in postgraduate education. China's Zhongyong Culture is of great significance for improving college students' mental health. Those with Zhongyong Practical Thinking can focus on the positive sides of stress events and strengthen their positive coping abilities, thereby alleviating depressive symptoms. Thus, cultivating the Chinese Zhongyong Thinking Mode can be part of a targeted psychological intervention (*He & Li, 2021*). Developing a special prevention program and cultivating postgraduates' positive coping styles can help students face challenges and setbacks with a positive and healthy mindset, and imparting the identified positive coping strategies to postgraduates can help them develop positive coping skills and be more flexible in dealing with issues. By doing so, they can avoid academic burnout and depressive symptoms and promote their well-being. Psychoeducation intervention in the academic environment may be an effective means to reduce academic distress. College students who have taken psychoeducation courses on evidence-based strategies for coping have lower academic distress levels and more positive perceptions of mental healthcare (*Savell et al., 2023*). A relevant study has also confirmed the intervention effect of Adlerian psychotherapy in reducing the academic burnout of college students (*Zheng, Wang & Shen, 2021*). In addition, assessing stress, burnout, and psychological health regularly is beneficial for both schools and students because it can inform educators of effective ways to identify students at risk and help students raise their awareness as well (*Yusoff, Hadie & Yasin, 2021*).

## Study limitations

This study also has some limitations. First of all, the participants were sampled from a single institution in China with a relatively small size, which can hardly represent the general postgraduate population, even though the sample size of 266 was sufficient for the path analyses (which require a sample size of more than 200 (*Huang, 2005*)). Thus, multi-center studies should be conducted to make the research findings generalizable to other studies. Secondly, the cross-sectional design could not deduce the causal relationship of mediating mechanisms. To fully understand these relationships, longitudinal studies should be carried out to verify causality. Thirdly, subjective biases cannot be ruled out because the collected data about depressive symptoms and related factors are self-reported. Finally, this study did not take into account academic factors that might make a difference in this model. Future studies may need to address this limitation.

## CONCLUSIONS

This study has established an optimal structural model to explore the relations between depressive symptoms, stressors, coping styles, and academic burnout among Chinese postgraduates. The findings indicate that stressors both directly and indirectly affect depressive symptoms through the simple mediating role of academic burnout, positive coping style, and negative coping style. Moreover, the findings of this study highlight the mediating roles of coping styles and academic burnout chain in the relationship between stressors and depressive symptoms among postgraduates. Therefore, to prevent and manage depressive symptoms among stressed postgraduates, interventions can be designed to cultivate their positive coping style and reduce academic burnout according to their mediation effects on the relationship. In the end, encouraging postgraduates to deal with stress positively may reduce the likelihood of the development of academic burnout and further alleviate depressive symptoms.

## ACKNOWLEDGEMENTS

We are grateful to all of the individuals for their involvement in the survey, including investigators and postgraduates for their support during data collection.

### Funding

This work was supported by a grant from the Science and Technology Department of Jilin Province, China (Grant Number: 20200101133FG). The funders had no role in study design, data collection and analysis, decision to publish, or preparation of the manuscript.

### Grant Disclosures

The following grant information was disclosed by the authors:
The Science and Technology Department of Jilin Province, China: 20200101133FG.

## Competing Interests

The authors declare there are no competing interests.

## Author Contributions

- Hong Shi conceived and designed the experiments, performed the experiments, analyzed the data, prepared figures and/or tables, authored or reviewed drafts of the article, and approved the final draft.
- Hanfang Zhao analyzed the data, prepared figures and/or tables, and approved the final draft.
- Minfu He analyzed the data, authored or reviewed drafts of the article, and approved the final draft.
- Zheng Ren analyzed the data, prepared figures and/or tables, and approved the final draft.
- Shixun Wang performed the experiments, prepared figures and/or tables, and approved the final draft.
- Li Cui performed the experiments, prepared figures and/or tables, and approved the final draft.
- Jieyu Zhao performed the experiments, prepared figures and/or tables, and approved the final draft.
- Wenjun Li performed the experiments, prepared figures and/or tables, and approved the final draft.
- Yachen Wei performed the experiments, prepared figures and/or tables, and approved the final draft.
- Wenjing Zhang performed the experiments, prepared figures and/or tables, and approved the final draft.
- Ziqiang Chen performed the experiments, prepared figures and/or tables, and approved the final draft.
- Hongjian Liu conceived and designed the experiments, authored or reviewed drafts of the article, and approved the final draft.
- Xiumin Zhang conceived and designed the experiments, authored or reviewed drafts of the article, and approved the final draft.

## Human Ethics

The following information was supplied relating to ethical approvals (i.e., approving body and any reference numbers):

This research was approved by the Research Ethical Committee of Jilin University School of Public Health.

## Data Availability

The data and code are available in the Supplemental Files.

## Supplemental Information

Supplemental information for this article can be found online at http://dx.doi.org/10.7717/peerj.16064#supplemental-information.

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
