# Peer review of "The mediating roles of coping styles and academic burnout in the relationship between stressors and depressive symptoms among Chinese postgraduates"

_PeerJ, doi:10.7717/peerj.16064_

## Round 0.1 · original submission · Major Revisions

Please address the reviewer's comments carefully, point by point.

Reviewer 1 ·

Basic reporting

no comment - please see section 4 - additional comments.

Experimental design

no comment - please see section 4 - additional comments.

Validity of the findings

no comment - please see section 4 - additional comments.

Additional comments

This work set out to test a model for explaining depressive symptoms in postgraduate students, based on perceived stress, coping strategies, and academic burnout. Though the goal seems worthy, there are several limitations to the manuscript that need to be revised. The most important of them is that the role that each variable play in the model is not made explicit and it is not, in my view, strongly substantiated. My main concerns are (1) that academic burnout be place as a process alike coping styles and (2) that stress is placed as predicting coping styles (in orders words, that individuals would adopt more or less of some coping styles if they felt more or less stressed), which I don’t think is accurate according to the literature in the area. There are other aspects that could be improved, namely an overall increased knowledge of literature about mental health of higher education students. Other than these more overarching comments, I provide specific comments below that I hope the authors will consider as constructive feedback.

Introduction
- The initial paragraph affirms both stress and depressive symptoms as prevalent in postgraduates. So, it sets the path for one or the other being the focus of concern. Perhaps it would be simpler to focus on depressive symptoms in this initial stage.
- The authors argue for the mediator role of coping strategies. This mediator role is theoretically placed between the stressor (events) and the outcomes (feeling stressed), unlike what is proposed in the model under scrutiny where coping styes are placed as mediators between feeling stressed and feeling depressed. The placing of coping styles between feeling stressed and feeling depressed would only make sense, in my opinion, if coping styles were thought of as a moderator; in order words, one would expect that the pathway linking stress to depressive symptoms be weaker when positive coping is high and be stronger when negative coping is high.
Method
- The authors used a measure of stressors to measure what they refer to as stress. This is confusing. I am not sure how this measure works. If I understood correctly, the respondent is asked to refer to the perceived impact several potential stressors might have had on their lives? So, this is a measure of perceived impact of stressors, with that impact being perceived as stress? I think it necessary to be more explicit about this. Please also note that even if you are assessing stressors, a moderation model still seems to be more applicable: if seems more plausible that the impact of stressors on depression be variant in the presence of various levels of coping strategies, rather than stressors predicting the diverse use of coping strategies and then impacting on depression.
- Strictly speaking, the authors are using path analysis (not SEM).
- It is necessary for the authors to refer to the adjustment indicators they considering when judging their model.
- Please inform on the software used for data analyses.
Results
- The average age presented in the text refers to the female participants only?
- Was there any data collected on academic factors (e.g., year of schooling, area of study etc.)? This would seem relevant as the authors’ assumption is that going through postgraduate training is stressful.
- There seems to be an overlap between the analyses: if there are significant correlation values there is likely to be significant regression values, and if there are significant regression values, there is likely to be a significant path analyses model. The regression adds some information when the authors consider the covariates, but the correlation analyses seem superfluous. The descriptive values on the variables of interest could be presented in the descriptive part of the results. Also, it would be helpful if the authors made these values more informative (e.g., are they close or not to values previously found for other samples?).
- On Table 3, why is the correlation value between positive and negative coping styles absent?
- Please provide p-values for paths shown in Figure 2 instead of referring to them in the text – of note, these are not correlation values; table 4 may be put into supplementary material.
- Please be explicit about the role that the dejection, improper behavior, and reduced personal accomplishment had in the model. Where they taken as measurements of academic burnout?
- Please avoid being repetitive – information that is described in detail in table 5 should not be described in the same detail in the text.
- Was the multivariate normality of the data explored prior to running the model? How was the model estimator chosen?
Discussion
- The authors discuss the multiple factors in the life of students that may stress them, related to their academic tasks. However, they collected no information on these factors of stress and so this discussion seems too overgeneralized. Alternatively, the authors could look into their instrument for assessing stress and look into the stressors that were referred to by their sample. If the indication that academic stress is overarching is true, then those stressors should be more obvious than others. The authors could then make the case for the presence and influence of academic burnout in their sample and their model.
- It is necessary to discuss the lack of a significant pathway between negative coping styles and depressive symptoms.
- The implications section would benefit from being more explicit and based on existing literature. What has been done to promote the mental health of college students? How can it be a steppingstone to address positive coping strategies and academic burnout in specific?
- How did the authors arrive at a required sample size of more than 200?
- Other limitations the authors might want to consider are that they did not consider academic variables that might make a difference in this model, or that they considered only self-reported data.
General comments
- The use of the English language needs to be revised. For example, see sentence starting at line 78.
- Please be consistent in using only two decimal places throughout the manuscript.

Reviewer 2 ·

Basic reporting

I believe the manuscript entitled "The mediating roles of coping styles and academic burnout in the relationship between stress and depressive symptoms among Chinese postgraduates: a structural equation modeling analysis" is written in a clear and professional manner. However, there are some areas that still need improvement. One weakness is the lack of proper citations and references for certain statements in the text. Additionally, the literature references are insufficient, with less than half of them being from the last 5 years, despite the availability of more recent articles on the subject. To address these issues, it is recommended to include citations in parts of the manuscript where information is provided without proper referencing, for example in the first two sentences of the introduction section. Additionally, the sentence "Chronic stress refers to stress that lasts for over a year" (lines 60-61) should be supported by a citation. Furthermore, in the discussion section, between lines 271-274, there are statements that require citations as they are not based on the results.
I suggest you double-check the data from the report on National Mental Health Development in China in lines 43-46 that indicate more than 100% together. As well the sentence in line 46 "Stress was prevalent in the group." is out of context and I could not understand which group is referring to.
The background section should be revised to incorporate new information and ensure smooth transitions between different variables and ideas. It is important to effectively link paragraphs when transitioning between topics. However, overall, the context of the manuscript is sufficient and relevant.
The raw data, figures, and tables in the manuscript are clear and well-described, which is commendable. The hypotheses are well defined and discussed.

Experimental design

The manuscript aligns with the aims and scope of the journal. The knowledge gap was identified, and the statements contribute to filling that gap. Regarding the methods adopted, they were well performed. However, it is suggested to provide more details regarding the approach to the participants, the timing of data collection, and the duration of the process. These details are crucial for reproducibility by other investigators.

Validity of the findings

The presentation of results is well done, and the discussion is important and relevant. The data provided are robust, statistically sound, and controlled, demonstrating the strength of the findings. In the conclusion section, I suggest exploring more this section by supporting the results and aim of the investigation. I also suggest including the term "chronic" in the sentence on line 315, specifically "especially for those under chronic stress."

---

## Round 0.2 · Minor Revisions

The reviewers have not responded to invitations to re-review this revision so I have checked all your responses and revision. Basically, the revised manuscript has improved quite a lot. But I found many places that need to redo. For example, when you describe "e.g." you need to a "," after period mark "." In addition, you need to check your English writing again. Though you said you have checked the previous manuscripts by Proofing English Studio, I suggest you ask an English expert to check them again.

**Language Note:** The Academic Editor has identified that the English language must be improved. PeerJ can provide language editing services - please contact us at copyediting@peerj.com for pricing (be sure to provide your manuscript number and title). Alternatively, you should make your own arrangements to improve the language quality and provide details in your response letter. – PeerJ Staff

---

## Round 0.3 · accepted · Accept

You have done an acceptable feature. However, there are still some minor editing problems. Congratulations.